# Wild-Mouse-Derived Gut Microbiome Transplantation in Laboratory Mice Partly Alleviates House-Dust-Mite-Induced Allergic Airway Inflammation

**DOI:** 10.3390/microorganisms12122499

**Published:** 2024-12-04

**Authors:** Md Zohorul Islam, Danica Jozipovic, Pablo Atienza Lopez, Lukasz Krych, Banny Silva Barbosa Correia, Hanne Christine Bertram, Axel Kornerup Hansen, Camilla Hartmann Friis Hansen

**Affiliations:** 1Department of Veterinary and Animal Sciences, Faculty of Health and Medical Sciences, University of Copenhagen, 1870 Frederiksberg, Denmarkakh@sund.ku.dk (A.K.H.); 2Section on Pathophysiology and Molecular Pharmacology, Joslin Diabetes Center, Boston, MA 02215, USA; 3Department of Microbiology, Harvard Medical School, Boston, MA 02115, USA; 4CSIRO Health & Biosecurity, Australian Centre for Disease Preparedness, Geelong, VIC 3220, Australia; 5Department of Food Science, Faculty of Science, University of Copenhagen, 1958 Frederiksberg, Denmark; 6Department of Food Science, Aarhus University, 8200 Aarhus, Denmark

**Keywords:** animal models, fecal microbiota transplantation, feral mice, gut microbiota, immunity, microbial metabolites, translatability, wild mice

## Abstract

Laboratory mice are instrumental for preclinical research but there are serious concerns that the use of a clean standardized environment for specific-pathogen-free (SPF) mice results in poor bench-to-bedside translation due to their immature immune system. The aim of the present study was to test the importance of the gut microbiota in wild vs. SPF mice for evaluating host immune responses in a house-dust-mite-induced allergic airway inflammation model without the influence of pathogens. The wild mouse microbiome reduced histopathological changes and TNF-α in the lungs and serum when transplanted to microbiota-depleted mice compared to mice transplanted with the microbiome from SPF mice. Moreover, the colonic gene expression of *Gata3* was significantly lower in the wild microbiome-associated mice, whereas *Muc1* was more highly expressed in both the ileum and colon. Intestinal microbiome and metabolomic analyses revealed distinct profiles associated with the wild-derived microbiome. The wild-mouse microbiome thus partly reduced sensitivity to house-dust-mite-induced allergic airway inflammation compared to the SPF mouse microbiome, and preclinical studies using this model should consider using both ‘dirty’ rewilded and SPF mice for testing new therapeutic compounds due to the significant effects of their respective microbiomes and derived metabolites on host immune responses.

## 1. Introduction

The introduction of procedures for generating clean specific-pathogen-free (SPF) mice has resulted in mice lacking the diverse microbiota commonly found in wild mice [1]. The SPF concept was introduced in the 1950s to avoid fatal and subclinical infections; both to standardize the environment of laboratory mice and to prevent the risk of zoonoses. As a result, founding breeders are now produced by cesarean section or embryo transfer, given the very limited microbiota in the form of the Altered Schädler Flora, and subsequently bred, maintained, and used for generations in protected barrier facilities [2]. This has led to a lack of microbial diversity and functional loss in the gut microbiome [3]. Consequently, laboratory-derived SPF mice are less suitable for mimicking natural microbiome functions [1,2] and have an immature immune system that resembles that of a newborn baby [4,5]. Hence, there are serious concerns that their clean environment results in poor bench-to-bedside translation in various animal models for human inflammatory diseases.

Recent studies have highlighted the potential of generating mice with a high resemblance to wild mice (i.e., rewilded mice) via different methods to improve immune responses and metabolic function. Cohousing with feral or pet shop mice has demonstrated the importance of pathogen exposure, and reconstitution of SPF mice with wild-mouse-derived gut microbiota has highlighted the necessity of the wild microbiome to stimulate proper immune and metabolic functions in mice [4,6,7]. The rewilded models recapitulate human immune responses better in preclinical studies where laboratory mice have failed to [4,6]; however, the extent to which this outcome is driven by pathogens and/or specific gut bacteria needs further investigation. The aim of this study was therefore to test whether pathogen-free wild-mouse-derived gut microbiota transplantation into SPF mice could change the immune response and sensitivity to disease in a commonly used inflammatory mouse model, the house dust mite (HDM)-induced allergic airway inflammation model, and to pinpoint the microbial functions potentially involved.

The animal model features many similarities to human asthma. Asthma is a chronic inflammatory disease of the respiratory system affecting more than 300 million people worldwide [8]. The pathogenesis of asthma is complex, with diverse clinical phenotypes, but eosinophilic or neutrophilic lung inflammation and the release of Th2-type inflammatory cytokines in an allergic response to environmental antigens are similar between this model and human disease [9,10]. While numerous host factors contribute to human asthma susceptibility, environmental factors such as lifestyle changes and early-life microbial exposures also play significant roles in asthma development [11]. Of particular interest, several studies have reported higher asthma rates in urbanized or industrialized populations than in their rural counterparts in various geographic regions [12,13,14]. Considering these observations, together with previous findings of exaggerated airway eosinophilia in ultraclean germ-free models of allergic airway inflammation [15], we expected the microbiomes of SPF and wild mice to differentially modulate the immune and inflammatory landscapes of the murine asthma model. We hypothesized that transplantation of a wild-mouse-derived gut microbiome would confer protection against allergic airway inflammation in laboratory SPF mice, even in the absence of pathogens, by restoring lost functions of the microbiome and enhancing immune regulation in the host.

## 2. Materials and Methods

### 2.1. Ethical Approval

This animal study was approved by the Animal Experimentation Inspectorate of the Ministry of Food, Fisheries, and Agriculture in Denmark, according to License No. 2017-15-0201-01262. The experiments were conducted following the Danish Animal Experimentation Act (Consolidation Act No. 474 of 15 May 2014) and the EU Directive 2010/63/EU on protecting animals used in scientific research.

### 2.2. Experimental Design

Male and female BALB/cJBomTac mice were purchased from Taconic Biosciences (Lille Skensved, Denmark) and bred in open-type 1284L Eurostandard II L cages (Techniplast, Varese, Italy) at the AAALAC-accredited and barrier-protected Frederiksberg campus’ experimental rodent facility at the Faculty of Health and Medical Sciences, University of Copenhagen, Denmark. At 15 days of pregnancy, the female mice were transferred to the AAALAC-accredited isolator facility in a hypopressure high-efficiency particulate air-ventilated flexible film isolator with 12 cages (PFI Systems, Milton Keynes, UK). Both facilities maintained a controlled, constant temperature of 22 ± 2 °C, a relative humidity of 45–55%, and a 12/12 light–dark cycle. All cages were supplied with aspen bedding material (Tapvei, Harjumaa, Estonia), aspen chew blocks, cotton nestles, a play tunnel, and a mouse igloo house (Brogaarden, Lynge, Denmark). The mice were fed ad libitum with an irradiated Altromin 1314 diet (Brogaarden) and sterile bottled water throughout the entire housing period. Health monitoring, under the guidance of the Federation of European Laboratory Animal Science Associations [16], revealed none of the listed infections.

At the time of moving the pregnant mice to the isolators, the mice were treated with a combination of 500 mg/L vancomycin, 1 g/L neomycin, and 1 g/L ampicillin in their drinking water for three consecutive days to deplete their intestinal microbiome. The pregnant BALB/c mice were randomly divided into two groups and housed in separate isolators. The two groups of pregnant mice were orally gavaged with either wild- or SPF-mouse-derived cecal contents to obtain pups colonized from birth with the respective microbiomes (group sizes were based on a minimum of three litters per group with a minimum of two pups per litter, to avoid experimental litter bias). The male and female pups (approximately 50:50) in each group were also given three doses of oral drops of cecal contents from the respective donor mice in the first three weeks after birth (Figure 1A). The wild (*n* = 13) and SPF (*n* = 8) microbiome-associated pups were weaned at 4 weeks of age, used for the HDM-induced asthma model starting at 5 weeks, and euthanized at 7 weeks of age (1 pup = 1 exp. unit). It was not possible to blind the main researcher during the experiment due to the separate isolators needed to separate the two groups. During euthanization, assisting researchers were blinded during the sampling and processing of the samples.

### 2.3. Cecal Microbiome Transplant

One wild female mouse (*Mus musculus*) was caught from a backyard garden in the area of Roskilde, Denmark, and was used as a wild microbiome donor. One adult female BALB/cJBomTac mouse from Taconic Biosciences (Lille Skensved, Denmark) was used as the SPF microbiome donor. The cecal contents for the microbiome transplant were prepared under aseptic conditions within an anaerobic biosafety cabinet. The donor mice were euthanized and immediately transferred to a biosafety cabinet. The cecum was longitudinally incised, and the entire cecal content was collected in a sterile 50-mL Falcon tube. The collected cecal contents were diluted at a 1:15 ratio (grams of cecal content to milliliters of cryoprotectant). A homogeneous slurry was prepared by gently vortexing and mixing the contents. To remove coarse particles and debris, the slurry was filtered through a 70-micrometer strainer. The filtered suspension was then aliquoted into 2 mL internally threaded plastic screw-capped cryovials and stored at −80 °C. To administer the donor microbiome to the recipient pregnant mice, flexible plastic tubing, oral gavage needles, and sterile 1 mL syringes were used. To initiate the transfer of the cryopreserved cecal contents, the samples were thawed in a 37 °C water bath until only a small piece of ice remained. Subsequently, 200 µL of cecal suspension was orally gavaged into each pregnant mouse as a single dose. Additionally, to maintain microbiome exposure, a few drops of cecal contents were given orally by dripping the pups into the mouth once a week during weeks 0, 1, and 2 after birth.

### 2.4. House-Dust-Mite-Induced Allergic Airway Inflammation

At five weeks of age, the mice were sensitized with HDM extract (Greer Laboratories, Inc., Lenoir, NC, USA), followed by subsequent challenge with the same allergen on days 7, 8, 9, 10, and 11 post-sensitization. On day 13, the mice were euthanized to observe any allergic airway inflammatory phenotypes. A 1 mg/mL HDM stock was prepared by mixing 4.57 mL of sterile PBS with 4.57 mg of HDM protein per vial of HDM extract. For sensitization, each mouse was provided 40 µg of sensitizing HDM solution (1 mg/mL), which required the administration of 40 µL of HDM stock per mouse. For the challenges, 10 µg of the HDM challenge solution per mouse per dose was needed. The challenge solution was prepared by mixing 10 µL of HDM stock solution with 30 µL of sterile PBS to prepare 40 µL of HDM challenge solution per mouse. To ensure proper sedation during the procedures, the mice were anesthetized via a subcutaneous injection of 150–200 µL of xylazine–ketamine mixture. The xylazine–ketamine mixture was prepared by combining 0.2 mL of ketamine (100 mg/mL) from MSD Animal Health (Rahway, NJ, USA), 0.25 mL of xylazine (20 mg/mL) from ScanVet Animal Health (Fredensborg, Denmark), and 1.55 mL of sterile water to make a 2 mL solution. The sensitization and challenge processes were conducted by carefully pipetting 40 µL of the appropriate HDM solution (sensitization/challenge) onto the nostrils of each mouse, one drop at a time, and allowing the mouse to inhale the solution effectively.

### 2.5. Mouse Sacrifice and Specimen Collection

The mice were anesthetized using Zoletil mix. A stock solution was made from Zoletil dry matter (25 mg/mL tiletamine + 25 mg/mL zolazepam) from Virbac (Carros, France), 10 mL of xylazin (20 mg/mL), and 0.5 mL of butorphanol (10 mg/mL) from Le Vet Beheer B.V. (Utrecht, The Netherlands), which was further diluted to prepare a ready-to-use mixture by diluting 2 mL of stock solution with 5.2 mL of saline and 2.8 mL of butorphanol. Once fully sedated with a 6.8 mL/kg body weight ready-to-use mixture, retroorbital blood was collected using microhematocrit capillary tubes and transferred to an Eppendorf tube. The blood samples were kept at room temperature for 30 min to allow for clot formation before being placed on ice, followed by centrifugation at 8000× *g* for 8 min to obtain clear serum at the end of the day. Sera samples were sent for serological testing of both the donors and the recipient pups at QM Diagnostics (AT Nijmegen, The Netherlands), and all the tested pathogens were negative (*Clostridium piliforme*, General parvovirus [rNS-1], Minute virus of mice, Mouse hepatitis virus, Mouse Parvovirus [rVP2], Mouse rotavirus/EDIM, Pneumonia virus of mice, Sendai virus, and Theiler’s encephalomyelitis virus [GD VII]). Fecal samples were collected from the recipient mice at euthanization and sent to IDEXX BioAnalytics (Kornwestheim, Germany) for PCR evaluation of parasites following their Protozoa and Pinworm PCR profile, which include tests for *Aspiculuris tetraptera*, *Chilomastix* spp., *Cryptosporidium* spp., *Eimeria* spp., *Entamoeba muris*, *Giardia muris*, *Spironucleus muris*, *Syphacia muris*, *Syphacia obvelata*, and *Tritrichomonas muris.*

Next, 0.1 mL of the Zoletil mix stock solution was subsequently injected into the mice, and to ensure euthanasia, the abdomen was opened and the diaphragm was punctured without damaging the lungs. Through a small incision in the trachea, the lungs were rinsed with cold sterile PBS using flexible plastic oral gavage needles, and the collected bronchoalveolar fluid (BAL) was transferred to a 15 mL Falcon tube for the subsequent counting of cells (i.e., primary read out) and further flow cytometric analysis. The entire middle right lobe of the lung was collected and frozen on dry ice for cytokine determination, and the inferior lobe of the lungs was collected in Carnoy’s solution for lung histology. For gene expression analysis, 1 cm of the ileum, closest to the cecum, was collected, and the lumen contents were removed. The tissue was then cut into small pieces and preserved in RNAlater. Similarly, 1 cm of the colon was collected after the sigmoid curve and kept in RNAlater for gene expression analysis. The cecal contents were collected in a separate Eppendorf tube and preserved at −80 °C for metabolomics analysis.

### 2.6. Flow Cytometry

The BAL fluid was weighed and centrifuged at 800× *g* for 5 min at 4 °C. The cell pellet was resuspended in 150 μL of PBS supplemented with a 1:100 CD16/CD32 monoclonal antibody (FcBlock, eBioscience, San Diego, CA, USA) before staining with an antibody cocktail containing anti-mouse CD11c (clone N418) labeled with AF488; MHC-II (clone M5/114.15.2) labeled with PE-Cy7; Siglec-F (clone S17007L) labeled with BV421; Ly-6G (clone 1A8) labeled with BV605; CD11b (clone M1/70) labeled with BV650; CD19 (clone eBio1D3) labeled with APC; and CD3 (clone 145-2C11) labeled with APC. Dead cells were excluded using PI staining. A pooled suspension was used for single stains, fluorescence minus one (FMO) controls, and unstained controls. The cells were washed twice with PBS and stained with 50 μL of the antibody mixture (all diluted 1:100 in FACS staining buffer, eBioscience) in the dark for 30 min. Following staining, the cells were washed twice with PBS, resuspended in 200 μL of PBS, run on a MACSQuant Analyzer 16 flow cytometer (Miltenyi Biotec, Bergisch-Gladbach, Germany), and analyzed with the included MacsQuantify software version 2.13.3.

### 2.7. Lung Histology

Lung sections were stained using hematoxylin and eosin (H&E) as well as periodic acid-Schiff (PAS) and subsequently examined under a microscope for overall inflammatory scoring and to determine mucus accumulation following previously established procedures [17]. Briefly, six random fields were selected from each slide using a 10× objective lens, with an emphasis on both bronchioles and blood vessels, and were scored blindly. The sections were graded on the following scale: 0 for no inflammation; 1 for occasional cuffing with inflammatory cells; 2 for most bronchi or vessels surrounded by a thin layer (one to five cells) of inflammatory cells; and 3 for most bronchi or vessels surrounded by a thick layer (more than five cells) of inflammatory cells. For the PAS-stained sections, the percentage of PAS-positive bronchi was calculated based on the number of PAS-positive bronchi and total bronchi in six random fields in each sample.

### 2.8. Cytokine Measurement

We measured the concentrations of cytokines in the middle right lung lobe and serum using a U-PLEX TH1/TH2 Combo (mouse) kit (Meso Scale Discovery, Rockville, MD, USA) containing IFN-γ, IL-1β, IL-2, IL-4, IL-5, IL-10, IL-12p70, TNF-a, CXCL-1, and IL-13. The test was performed following the manufacturer’s guidelines. The plate was read on a MESO QuickPlex SQ 120 instrument and analyzed with the included software Discovery Workbench 4.0 (Meso Scale Discovery).

### 2.9. ELISA for IgG and IgE Antibodies

Serum total IgE concentrations were measured at euthanasia using a Mouse IgE ELISA Kit (Bethyl Laboratories, Montgomery, TX, USA) as previously described [18]. The serum was diluted to a ratio of 1:20, and subsequently, 100 μL of the diluted sample was loaded onto a 96-well ELISA plate, along with the test standard. The ELISA procedure was performed according to the manufacturer’s instructions, and the absorbance of the plate was read at a wavelength of 450 nm on a PowerWave X microplate spectrophotometer (KC4 v3.4, Rev 21; Bio-Tek Instruments, Inc., Winooski, VT, USA).

### 2.10. Gene Expression Analysis by qPCR

In total, 1 cm of ileum was collected in RNAlater and stored at −80 °C for subsequent gene expression analysis of gut-barrier-related genes. Homogenization, RNA isolation with a MagMAX-96 RNA Isolation Kit (Ambion, Kaufungen, Germany), and cDNA synthesis using a High-Capacity cDNA Reverse Transcriptase Kit (Applied Biosystems, Foster City, CA, USA) were performed as described previously. Quantitative PCR (qPCR) of genes associated with gut barrier functions, *Muc1*, *Muc2*, *Ocln*, *Tjp1*, *Cldn8*, *Cldn15*, *Reg3g*, and *Pla2a2g*, as well as genes associated with immune regulation, *Cd8a*, *Ifng*, *Tgfb*, *Tnf*, *Rorc, Il4*, *Il5*, *Il12b*, *Gzmb*, and *Gata3*, was performed using TaqMan gene expression assays (Applied Biosystems) on a Bio-Rad C1000 Touch CFX96 Real-Time System Thermal Cycler (Bio-Rad, Hercules, CA, USA). *Hprt* was used as a reference gene. The amplification data were analyzed using the accompanying CFX Maestro software version 2.3 (Bio-Rad) to obtain threshold cycle (Ct) values. GenEx 6 (MultiD Analyses AB, Gothenburg, Sweden) was used for qPCR data transformation. The Genorm and NormFinder methods in GenEx were used to validate the stability of the selected reference genes, and the Ct values were normalized to the reference gene. For each gene, normalized expression levels were set relative to the sample with the lowest expression to establish relative quantities (RQ) and were log2-transformed before statistical testing. The false discovery rate was used to correct for multiple comparisons.

### 2.11. Serum Lipopolysaccharide (LPS)

The serum LPS concentration at euthanization was measured using the Lonza Pyrogene Recombinant Factor C Endotoxin Detection Assay (Lonza, Basel, Switzerland). Sera samples were diluted 1:100, and procedures were carried out following the manufacturer’s instructions. The absorbance of the plates was measured at time zero and after one hour of incubation at 37 °C with catalyzing reagents. Concentrations were calculated as the delta value of the two readings.

### 2.12. High-Throughput Sequencing of the Gut Microbiota

Fecal samples were collected at 4 weeks of age before model induction. DNA was extracted using a Bead-Beat Micro AX Gravity Kit (A&A Biotechnology, Gdynia, Poland) according to the manufacturer’s instructions. The relative abundance of the bacterial community was evaluated using nanopore-based sequencing of the nearly full 16S rRNA gene amplicon via GridIONx5 (Oxford Nanopore Technologies, Oxford, UK) as previously described [19]. The microbiome abundance table was used for subsequent statistical analysis employing Microbiome Analyst [20]. A total of 422 OTUs were detected from 471,435 sequencing reads (with an average count per sample of 14,732). Prior to conducting any statistical analysis, the data were filtered based on a low filter count at a 10% prevalence threshold. Subsequently, the data were normalized by scaling them using the total sum scaling method. First, Chao1 alpha diversity was measured to determine within-sample diversity and was compared between experimental groups using a paired t test. Next, between-sample diversity (beta diversity) was calculated using the Bray–Curtis dissimilarity method at the feature level, and group comparisons were tested using the pairwise permutational MANOVA (PERMANOVA) method. Finally, a heatmap and clustering were generated at the genus level using the Euclidean distance measure and the Ward clustering algorithm.

### 2.13. Mycobiome Analysis

The mycobiome analysis was carried out from the same extracted DNA that was used to analyze the microbiome; however, in this case, specific primers that targeted the ITS region were used (ITS1 forward and ITS4 reverse) [21]. The mycobiome abundance table was obtained by adapting LACA [22] to the expected size of the regions sequenced (300–2500 bp) using the UNITE fungal ITS-specific database [23]. All plots and post hoc analyses were conducted using R.

### 2.14. Metabolomics Analysis

Cecal samples were thawed on ice, and 100 mg of the sample material (wet weight) was mixed with 0.2 mL of distilled water by vortexing for 20 s. Samples were subsequently centrifuged for 10 min at 4 °C and 14,000× *g*, 0.5 mL of supernatant was collected, and the pH was measured using a pH meter fitted with a silver electrode (Radiometer, Copenhagen, Denmark). A volume of 0.6 mL of deuterium oxide phosphate buffer (0.1 M, Pd = 7.4) was added, and the mixture was vortexed for 20 s and centrifuged under the same conditions. A 0.5 mL aliquot of the supernatant was filtered through 0.5 mL of 10k Millipore centrifugal filter units by centrifugation for 30 min at 4 °C and 14,000× *g*, after which, 0.4 mL of the supernatant was collected. Finally, the supernatant was transferred to a 5 mm NMR tube and 0.2 mL of deuterium oxide containing 3-(trimethylsilyl)-propionic-2,2,3,3-d4 acid and 0.0075% sodium salt (TSP d4) was added [24].

NMR spectroscopy was conducted at 300 K on a 14 T Bruker Avance III spectrometer (Bruker BioSpin, Rheinstetten, Germany) equipped with a BBI 5 mm probe with gradients; an automated tuning and matching accessory (ATMATM); a BCU-I for the regulation of temperature; and a SampleJet robot cooling system set to 5 °C as a sample exchanger. Proton NMR spectra were acquired using a NOESY presaturation pulse sequence (Bruker 1D noesygppr1d pulse sequence), with 64 K data points, a spectral width of 20 ppm, an acquisition time of 2.75 s, 4 s of dummy scans, a relaxation delay of 4 s, 64 scans, and a fixed receiver gain. The free induction decays (FIDs) were multiplied by a 0.3 Hz exponential function prior to Fourier transformation. Phase and baseline corrections were carried out, the reference standard TSP-d4 signal was adjusted to δ 0.00, and the spectra were assigned using Chenomx database values. For quantification, Chenomx software 7.2 was used to integrate and quantify the metabolite peaks relative to the TSP-d4 standard.

Multivariate analysis was carried out using SIMCA software v.17. Principal component analysis (PCA) and orthogonal partial least squares–discriminant analysis (OPLS-DA) were also conducted on the data that were preprocessed by autoscaling, and 3 principal components were used. Hierarchical clustering heatmaps were generated using the MetaboAnalyst 5.0 platform (http://www.metaboanalyst.ca/faces/home.xhtml (accessed on 1 November 2022)) with Euclidean distance metrics and Ward’s clustering algorithm. Metabolic pathway analysis was performed by applying the differentially abundant metabolites cross-listed with the pathways in the Kyoto Encyclopedia of Genes and Genomes (KEGG)—using the pathways previously identified in the Kyoto Encyclopedia of Genes and Genomes (KEGG) of *Rattus norvegicus*—and the top altered pathways were identified and built according to the potential functional analysis. For this purpose, the MetaboAnalyst 5.0 platform was also used.

### 2.15. Statistics

GraphPad Prism version 9 (GraphPad Software, Boston, MA, USA) was used for statistical analysis; *p* values < 0.05 were considered significant but tendencies with *p* values < 0.1 are also shown. Differences between the two groups were estimated by an unpaired two-tailed Student’s *t* test or the Mann–Whitney test if the data did not follow a Gaussian distribution. Welch’s correction was included in the *t* test if variances were unequal according to the *F* test.

## 3. Results

### 3.1. Wild-Mouse-Derived Gut Microbiota Reduces Allergic Airway Inflammation in Transplanted Mice

The diverse microbiomes of wild mice are lacking in laboratory SPF mice due to their clean barrier-restricted housing conditions. To test the importance of this on immune responses in a commonly used inflammatory disease model for asthma, we transplanted cecal content from a wild mouse into BALB/c mice and compared their sensitivity to house-dust-mite-induced airway inflammation with that of BALB/c mice with a ‘normal’ SPF mouse microbiome (Figure 1A). Interestingly, we observed that the wild-microbiome-associated mice weighed less than the SPF-microbiota-associated mice at the time of euthanasia (Figure 1B). There were also significantly fewer leukocytes in the BAL fluid collected at euthanasia (Figure 1C), which indicated less inflammation in the lungs of the wild-microbiome-associated mice. However, we were not able to detect a significantly decreased number of eosinophils specifically (Figure 1D), which is a hallmark of allergic airway inflammation induced in mice. The numbers of neutrophils, T cells, and B cells were also unaltered (Figure 1E–G), except for a tendency for fewer B cells in the wild-microbiome-associated mice (Figure 1G). We therefore measured cytokine levels in the lungs and serum, and while the levels of the classical type 1 and 2 proinflammatory cytokines and the anti-inflammatory cytokine IL-10 were similar between the groups (Appendix A), the level of proinflammatory TNF-α was significantly lower in the lungs of the wild-microbiome-associated mice than in those of the SPF-microbiome-associated mice (Figure 1H). This was verified in the sera, where TNF-α was similarly reduced in the wild-microbiome-associated mice (Figure 1I). The type 2 immune response is also characterized by strong IgE induction but no difference was observed in the serum IgE levels between the groups (Figure 1J). To further investigate the inflammatory state of the lungs, histopathological analysis was performed. The overall inflammatory score indicating the degree of infiltrating leukocytes in the lungs tended to be lower in the wild-microbiome-associated mice than in the SPF-microbiome-associated mice according to the BAL counts (Figure 1K–M); moreover, a strong tendency toward fewer mucin-positive bronchi was observed in the wild-microbiome-associated mice than in the SPF-microbiome-associated mice (Figure 1N–P). Overall, the observed differences in body weight, BAL cell counts, local and systemic TNF-α levels, and histopathological changes collectively support the hypothesis that wild gut microbiota can, to a minor degree, reduce susceptibility to allergic inflammation in the lungs of laboratory mice, even though the classical type 2 immune response was unaltered.

### 3.2. Wild-Derived Microbiome Transplantation Significantly Upregulated Intestinal Gene Expression Associated with Improved Gut Barrier Function

Intestinal barrier dysfunction is a pathological manifestation that seems to be a characteristic of potential underlying mechanisms in the gut–lung axis that can lead to airway inflammation. Studies have demonstrated impaired barrier function in individuals with bronchial asthma [25]. Moreover, wild mice have a microbiota-dependent impenetrable mucus layer in the colon compared to SPF mice [26]. To determine the effect of transplanting the wild-derived microbiome into SPF-born recipient mice on gut barrier function, we assessed the expression of specific genes related to intestinal integrity and immune function in the gut tissues. Notably, the expression of the *Muc1* gene, which encodes transmembrane mucin proteins involved in the first line of defense of the intestinal barrier, was significantly greater in both the colon and ileum of the wild-microbiome-associated mice than in those of the SPF-microbiome-associated mice (Figure 2A,B). Moreover, the gene expression of the tight junction protein occludin was upregulated in the wild-microbiome-associated mice, suggesting improved barrier function in the colon (Figure 2A) but downregulated in the ileum (Figure 2B). Only in the colon did we observe significantly reduced expression of the transcription factor *Gata3*—which promotes Th2 responses—in the wild-microbiome-associated mice compared to the SPF-microbiome-associated mice (Figure 2A). The serum endotoxin (LPS) concentrations were similar between the two groups (Appendix A), indicating comparable intestinal permeability in the two groups.

### 3.3. Distinct Microbiome Taxa Were Found Between Wild and SPF Recipient Mice

Next, we sought to identify specific microbial taxa and metabolites that were differentially abundant in the wild and SPF microbiomes and may have caused the anti-inflammatory effects observed in the recipient mice. We did not observe any significant changes in the overall alpha diversity between the wild- and the SPF-microbiome-associated mothers or pups at weaning (Figure 3A); however, the two recipient groups clustered separately in the principal coordinates analysis plot (Figure 3B). The microbiomes of both the mothers and the pups were significantly different (Figure 3C). After conducting a multivariable linear regression analysis using MaAsLin2 [27], we detected a significant difference in the abundance of seven taxa between the wild- and SPF-microbiome-associated pups at weaning. *Helicobacter* spp. (4.8 ± 0.8 (log2FC ± SE); *p* = 3.0 × 10^−6^; FDR = 9.4 × 10^−5^), *Desulfovibrio* spp. (3.7 ± 0.8 (log2FC ± SE); *p* = 0.0001; FDR = 0.002), *Alistipes* spp. (2.7 ± 1.1 (log2FC ± SE); *p* = 0.017; FDR = 0.087), and *Bifidobacterium longum* (1.3 ± 0.6 (log2FC ± SE); *p* = 0.038; FDR = 0.088) were significantly more abundant in the wild-microbiome-associated mice (Figure 3D). *Phocaeicola vulgatus* (−3.1 ± 0.9 (log2FC ± SE); *p* = 0.002; FDR = 0.018), *Ligilactobacillus murinus* (−1.6 ± 0.5 (log2FC ± SE); *p* = 0.006; FDR = 0.027), and *Akkermansia muciniphila* (−2.0 ± 0.9 (log2FC ± SE); *p* = 0.036; FDR = 0.088) were significantly more abundant in the SPF-microbiome-associated mice (Figure 3D). *Helicobacter* spp., *Desulfovibrio* spp., and *Bifidobacterium longum* were also clearly more abundant in the wild mouse donor cecum sample than in the SPF mouse donor cecum sample, whereas the differential abundances of *Alistipes* spp., *Phocaeicola vulgatus*, *Ligilactobacillus murinus*, and *Akkermansia* in the pups were not evident in the donor samples (Figure 3D). In contrast, a few additional genera such as *Faecalibacterium* spp. and *Erysipelatoclostridium* spp., which were clearly more abundant in the wild donor cecum sample than in the SPF donor sample, remained similar in the recipient groups (Figure 3D). The same differences observed in the pups could also not be found in the mothers, in which only *Klebsiella* spp. (1.1 ± 0.2 (log2FC ± SE); *p* = 2.3 × 10^−6^; FDR = 8.4 × 10^−5^) were significantly more abundant in the wild-microbiome-associated mothers (Figure 3D).

### 3.4. Presence of Intestinal Fungi, Protozoa, and Helminths Were Similar Between the Wild- and SPF-Microbiome-Associated Mice

In order to assess whether the differences in the phenotype were truly related to the intestinal bacteria, the full spectrum of the intestinal microbial alterations was analyzed. Fecal samples were PCR tested by IDEXX BioAnalytics Protozoa and Pinworm PCR profile but none of the tested parasites were detected in either group. Moreover, sequencing the ITS region of the fecal DNA was performed to analyze the mycobiome; however, PERMANOVA analysis on the robust Aitchison distance between fecal samples did not reveal any differences between the two groups (Figure 3E).

### 3.5. The Wild- and SPF-Mouse-Derived Microbiomes Induced Distinct Intestinal Metabolite Profiles in the Recipient Mice

Finally, the impact of wild-derived microbiome transplantation on cecal metabolites was investigated. In total, 51 metabolites were identified (Appendix A). Untargeted metabolomics analysis revealed a clear separation in the score plot between the wild- and SPF-microbiome-associated mice along the t (scores) axis (Figure 4A). These findings suggest that the wild-mouse-derived microbiome induces a distinct cecal metabolome, leading to unique metabolic activities in the gut of wild-microbiome-associated mice. A loading plot was generated to determine the contributions of specific metabolites to the observed separation (Figure 4B), which showed that fumarate and gallate were the most important metabolites for discriminating the two groups. Fumarate concentrations were greater and gallate concentrations were lower in wild-microbiome-associated mice than in SPF-microbiome-associated mice. An overview of the model validation of the OPLS-DA model is shown in Appendix A. Furthermore, univariate regression analysis was performed to detect significant differences in the cecal metabolome. The heatmap displays the overall concentration and distribution of different metabolites across samples (Figure 4C). The concentrations of twelve metabolites were significantly different between the wild- and SPF-microbiome-associated mice, including the concentration of the organic acid fumarate, which was significantly greater in the wild-microbiome-associated mice than in the SPF-microbiome-associated mice. In contrast, the concentrations of eleven other metabolites were significantly greater in SPF-microbiome-associated mice (Figure 4D). These eleven metabolites included one phenolic acid (gallate), one organic acid (4-hydroxyphenylacetate), and nine amino acids (aspartate, leucine, lysine, methionine, phenylalanine, threonine, tryptophan, tyrosine, and valine).

Analysis of the metabolic pathways associated with the metabolites found in the cecal content revealed alterations in four metabolic pathways when the two groups were compared, as follows, according to the order of the highest impact: citrate cycle, tyrosine metabolism, tryptophan metabolism, and arginine biosynthesis (Appendix A).

## 4. Discussion

There is an urgent need to address the poor validity of animal models. While animal models are essential for research and drug development, invalid animal models waste time and resources in the battle against rapidly growing epidemics of serious chronic diseases. In particular, the inability to reproduce results between animal experiments and confirm results in humans is one of the largest roadblocks in drug development [28]. Dirty mouse models could be one way to overcome the poor bench-to-bedside translation in preclinical research but ethical and safety issues exist with reintroducing wild life pathogens into our SPF facilities. We hypothesized that reintroduction of diverse microbiota from wild mice living in a natural habitat would be sufficient to safely normalize the immune response and reduce disease sensitivity in an HDM-induced allergic airway inflammation model. Wild-mouse-derived gut microbiome transplantation reduced lung inflammation and decreased the level of proinflammatory TNF-α in the lungs and serum of the recipient mice compared to those with the microbiome from SPF mice. These findings were accompanied by reduced colonic expression of the Th2 transcription factor *Gata3*, while the expression of other specific Th2 immune markers was comparable between the groups. The HDM-induced asthma model is acute with only five challenges, which could explain why changes in acute inflammatory responses in the lung were not accompanied by changes in adaptive immune parameters, such as IgE and classical type 2 cytokine levels, which are otherwise known to be reduced by a diverse microbiome early in life [29,30].

The protective effect of the wild microbiome compared to the microbiome from SPF mice living in abnormally clean environments is in line with the hygiene hypothesis stating that the shift from traditional rural lifestyles to modern industrialized environments has brought about significant changes in dietary habits, as well as increased sanitation practices, birth modes, increased use of antibiotics, vaccinations, etc., that have been associated with reduced microbial diversity and substantial alterations in the composition of the gut microbiome in urban populations [31,32,33,34]. The depletion of beneficial microbial communities resulting from these lifestyle changes has been linked to dysregulated immune functions and the escalating incidence of chronic inflammatory diseases in modern society [31,35]. Exposure to farm-derived bacteria or bacterial endotoxins also protects mice from developing an allergic response to ovalbumin or HDMs [36,37]. Our study is also in line with other mouse studies showing that a lack of microbial exposure exacerbates allergen-driven inflammatory responses in the lungs. Housing laboratory mice in clean high-barrier facilities aggravated lung immunity and caused more severe allergic lung disease in the HDM-induced model than in low-barrier facilities [38]. Cohousing with pet-shop mice also impairs lung eosinophil and innate lymphoid cell group 2 responses to intranasal allergen exposure, though only transiently [39]. Several mouse studies thus support the hygiene hypothesis, pointing toward the importance of the gut microbiota in regulating allergic airway responses to foreign allergens [40,41,42].

In contrast, a more recent study revealed no protection against HDM-induced allergies in wildling mice exposed to lifelong fulminant microbiota from wild mice [43]. The wildling mice developed robust Th2 cell responses. Furthermore, heightened levels of circulating IgE and IgG were found in adult wildling mice compared with SPF mice [43]; this phenomenon has also been previously observed in wild-caught mice [44,45], whereas cohousing SPF mice with pet store mice dampened humoral immune responses [46,47]. Discrepancies between studies comparing clean SPF with ‘dirty’ or rewilded mouse models may result from inconsistencies in generating rewilded mice and in the study design, as reviewed by Yeh et al. [48]. Differences in exposure to infections, fungi, and nematodes are relevant cofounders, as are differences in mouse strains and gut microbiota compositions, which make comparisons within and between experiments challenging. Both the donor and the recipient mice in our study were free of viruses, intestinal helminths, and protozoa, and the mycobiome was similar between the groups. Hence, it is important to stress that only the origin of the gut microbiota transplant was different between the groups in our study, whereas several of the other mentioned studies also implemented naturalized housing conditions for the ‘dirty’ rewilded mice as an alternative to traditional cage housing for laboratory rodents. The impact of housing alone on relevant immunological and physiological parameters is thus highly relevant to investigate further.

We could not confirm an overall reduction in the alpha diversity of the gut microbiota in SPF-microbiome-associated mice as we would have expected but the abundance of several important immunomodulatory genera was significantly altered. For instance, *Helicobacter* species have been associated with various effects on the host. Most species within the *Helicobacter* genus are pathogenic and linked to gastrointestinal disorders [49], which is the reason that these bacteria are actively eradicated by commercial mouse breeders. Additionally, in the present study, the SPF mice were declared free of *Helicobacter* in the health report; thus, in alignment with previous findings, the wild mice, not surprisingly, harbor more *Helicobacter* in their intestine than SPF mice [3]. Interestingly, *Helicobacter* is inversely associated with childhood asthma [50,51] and can prevent allergic asthma in experimental mouse models [52]. The wild microbiome also contained a greater abundance of *Alistipes* and *Bifidobacterium longum*, which is relevant because lower abundances of both *Alistipes* and *Bifidobacterium* are associated with an increased risk of asthma in human infants [53]. In addition, *Desulfovibrio* was enriched in the wild-microbiome-associated mice, which is a sulfate-reducing bacterium linked to alterations in gut barrier function and has been observed in various gastrointestinal disorders, such as inflammatory bowel disease [54]. Despite the pathogenic potential of those genera being more abundant in the wild microbiome, the wild microbiome appeared to have a beneficial impact on mucosal barrier function in the intestines, unless the high *Muc1* expression in the wild-microbiome-associated mice was, in fact, a host response mounted to prevent those microorganisms from adhering and penetrating through the intestinal wall. In support of this notion, no difference was observed in the serum endotoxin levels, indicating that intestinal permeability was independent of the origin of the microbiome transplant. In another study, feralizing laboratory mice in farmyard-type environments did not change intestinal mucus thickness and penetrability, despite a higher expression of genes encoding mucus components [55].

The differences in the cecal metabolome between the two recipient groups resulted in significant differences in pathways related to the citric acid cycle, arginine biosynthesis, and tyrosine and tryptophan metabolism. Interestingly, tryptophan metabolism in the gut has been linked to immune homeostasis and proposed to influence asthma pathophysiology [56], which may explain why wild-microbiome-associated mice were less sensitive to HDM-induced allergic airway inflammation. Furthermore, we observed significant elevations in the branched-chain amino acids (BCAAs) leucine and valine in the SPF-microbiome-associated mice than in the wild-microbiome-associated mice. BCAAs are known to be important immunoregulators [57], and previous studies have indicated that the metabolism and levels of BCAAs vary between the plasma of healthy controls and asthmatic patients [58] as well as in the urine [59]. Our findings highlight the importance of microbiome-driven metabolic alterations in shaping the overall immune landscape in wild versus SPF mice but understanding how specific metabolites are involved and their functional roles requires further investigation. Another limitation of this study is that it would be beneficial to repeat the study with additional SPF and wild FMT donors for reproducibility purposes, but due to the extent of running isolator projects, this has not been feasible. However, the same differences in gut microbiota between wild and SPF mice have been shown before in multiple studies, and the results would therefore be expected to be reproducible with additional donors.

Another limitation of this study is that it was only performed once, as it would be beneficial to repeat the study with additional SPF and wild FMT donors for reproducibility purposes. Nonetheless, similar differences in gut microbiota between wild and SPF mice have been shown before in other studies. It is also a limitation that isolator experiments have potential cage effects due to the completely separated groups in the different units. Finally, it is a limitation that vehicle-treated groups were not included as it would be relevant to test whether differences in, e.g., intestinal gene expression were induced by the microbiome directly or as an effect of the altered disease state.

## 5. Conclusions

In conclusion, our findings suggest that wild-derived mouse gut microbiome transplantation modulates the host’s intestinal mucus layer properties without affecting the intestinal permeability, as well as modulating the immune response and susceptibility to allergic inflammation in the lungs, possibly driven by changes in metabolites related to tryptophan metabolism. Preclinical studies using the HDM-induced allergic airway inflammation model should thus consider using both ‘dirty’ rewilded mice transplanted with a wild mouse microbiome and regular SPF mice for testing new therapeutic compounds.

Specific comparisons of immune responses in different rewilded models are continuously important to determine which microbial and pathogenic exposures are needed to best recapitulate different aspects of human diseases. Understanding the mechanisms underlying these effects on the host could pave the way for novel therapeutic strategies targeting the microbiome for the prevention and management of chronic inflammatory diseases such as asthma. Further research is thus warranted to elucidate the specific microbial taxa, metabolites, and pathways driving the observed effects in our study and to explore their translational potential in human health and disease.

## Figures and Tables

**Figure 1 microorganisms-12-02499-f001:**
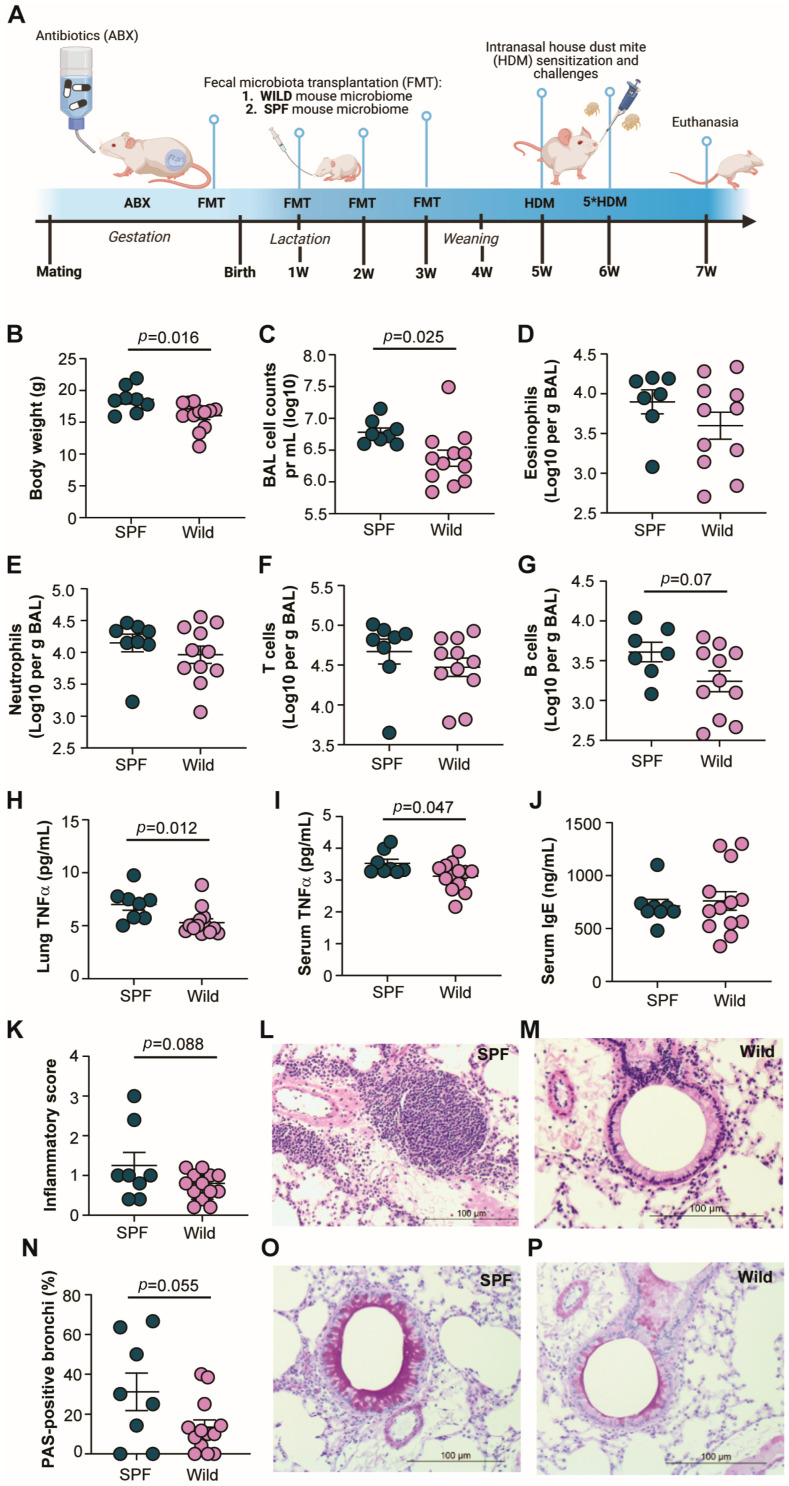
Compared with SPF-microbiome-associated mice, wild-mouse microbiome-associated mice showed reduced allergic airway inflammation: (**A**) Experimental design showing the timeline of antibiotic treatment (ABX) and microbiome transplantation (FMT) as well as house dust mite (HDM) sensitization and challenges before euthanization of male and female wild mice (wild) and specific pathogen-free (SPF) microbiome-associated BALB/c mice; (**B**) body weight; (**C**) bronchoalveolar fluid (BAL) cell counts; (**D**) numbers of eosinophils (CD11c-CD3-CD19-CD11b+Ly6G-SigF+) in the BAL fluid; (**E**) numbers of neutrophils (CD11c-CD3-CD19-CD11b+Ly6G+) in the BAL fluid; (**F**) numbers of T cells (CD11c-CD3+MHCII-) in the BAL fluid; (**G**) numbers of B cells (CD11c-CD19+MHCII+) in the BAL fluid; (**H**) lung concentration of TNF-α; (**I**) serum concentration of TNF-α; (**J**) serum total IgE concentrations; (**K**) histopathological inflammatory score in lung tissue, including representative images of H&E-stained sections (**L**,**M**); (**N**) percentage of mucin-positive bronchi, including representative images of PAS-stained sections (**O**,**P**), are shown for the wild (*n* = 13) and SPF (*n* = 8) microbiome-associated mice when euthanized at 7 weeks of age. All *p* values are given (significance < 0.05). The mean and SEM are shown.

**Figure 2 microorganisms-12-02499-f002:**
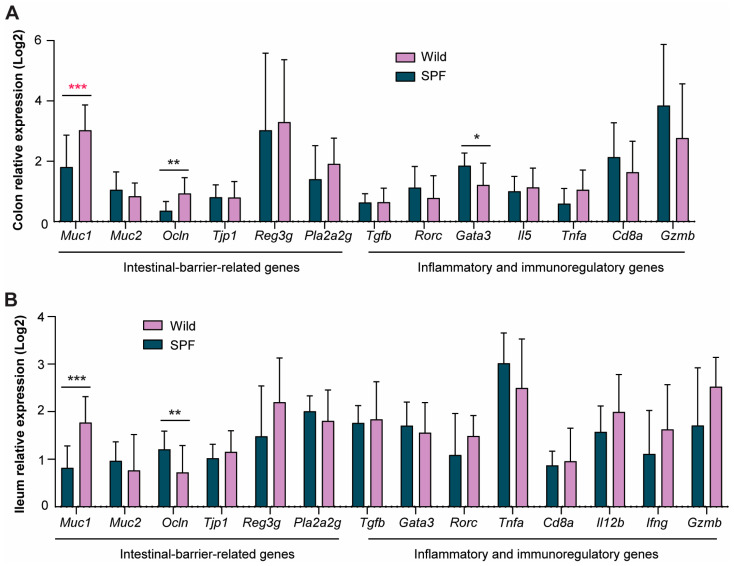
Transplantation of wild-mouse-derived microbiome resulted in significantly increased expression of the *Muc1* gene, which is related to mucosal barrier function. qPCR expression analysis of gut-barrier-related and immunoregulatory genes in the colon (**A**) and ileum (**B**) of wild (*n* = 13) and SPF (*n* = 8) microbiome-associated male and female BALB/c mice at 7 weeks of age after induction of HDM-induced allergic airway inflammation. * indicates *p* < 0.05, ** indicates *p* < 0.01, and *** indicates *p* < 0.001. Red stars indicate significance after FDR correction. The mean and SD are shown.

**Figure 3 microorganisms-12-02499-f003:**
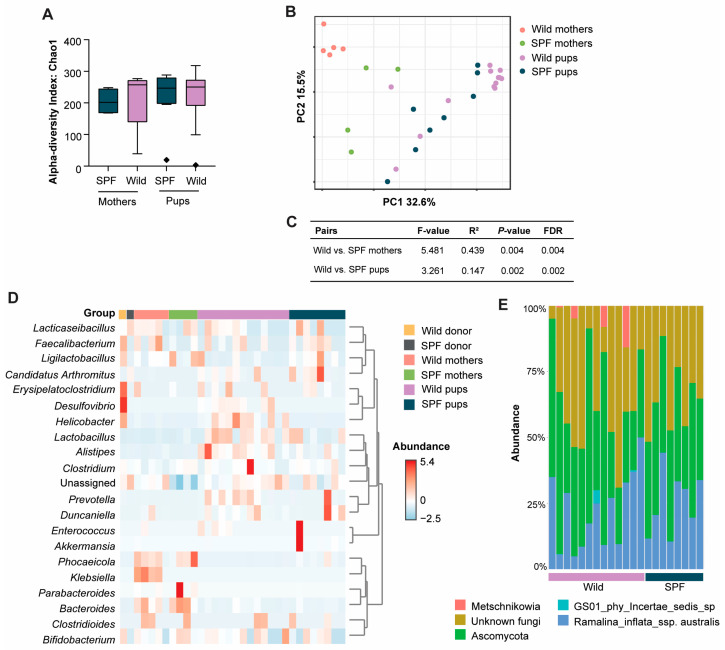
Differential abundance analysis revealed distinct microbiome taxa, but not mycobiome, between wild- and SPF-microbiome-associated mice: (**A**) Chao1 alpha diversity analysis depicting the fecal microbiome population diversity among the wild- and SPF-microbiome-associated pregnant mothers at birth (wild, *n* = 5 and SPF, *n* = 4) and their male and female pups at weaning (wild, *n* = 13 and SPF, *n* = 8); (**B**) principal coordinates analysis plot of the 16S rRNA gene amplicon sequencing of fecal samples from mothers and pups based on the Bray–Curtis dissimilarity method at the feature level; (**C**) pairwise permutational MANOVA (PERMANOVA) test results showing the differences in beta diversity between the groups. The F value (referring to the test statistic that measures the ratio of the variance between group means to the variance within the groups), the R^2^ value (representing the proportion of total variation in the data that is explained by the grouping variable), *p* values, and FDR-corrected *p* values are given for comparisons of the beta diversity observed in (**B**) of wild- and SPF-microbiome-associated mothers and pups as indicated; (**D**) heatmap visualizing the relative abundance of the top genera within each mouse fecal sample as well as in the donor cecum samples. The dendrogram cluster on the right side of the heatmap shows the abundance pattern of particular genera between sample groups; (**E**) bar chart showing the relative abundance of detected mycobiome taxa by ITS sequencing.

**Figure 4 microorganisms-12-02499-f004:**
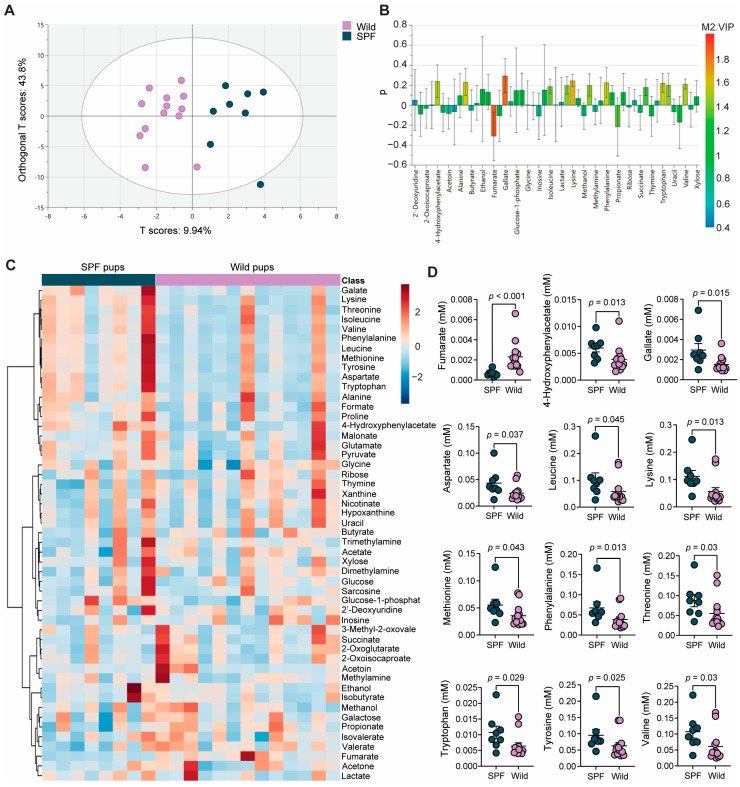
The wild and SPF microbiomes induced distinct metabolomic profiles in the cecal contents of the recipient mice: (**A**) score plot visualizing the distribution of samples based on their scores on the OPLS-DA components; (**B**) loadings plot showing the variable of importance for the projection (VIP) in component 1; (**C**) heatmaps showing differential levels of cecal metabolites between wild- and SPF-microbiome-associated pups at euthanization after HDM-induced airway inflammation; (**D**) differential concentrations of metabolites between wild- and SPF-microbiome-associated mice. An overview of the OPLS-DA model is shown in Appendix A. All *p* values are given (significance < 0.05). The mean and SEM are shown.

## Data Availability

The datasets generated and/or analyzed during the current study are available in the NCBI repository, project accession number PRJEB72838, (https://www.ebi.ac.uk/ena/browser/view/PRJEB72838 (accessed on 8 February 2024)), and in the Mendeley repository as DOI: 10.17632/48m83zmwys.1 (https://eur02.safelinks.protection.outlook.com/?url=https%3A%2F%2Fdata.mendeley.com%2Fdatasets%2F48m83zmwys%2F1&data=05%7C02%7Ccamfriis%40sund.ku.dk%7Cb6f23220de49464a33fa08dc57ec3971%7Ca3927f91cda14696af898c9f1ceffa91%7C0%7C0%7C638481919304815477%7CUnknown%7CTWFpbGZsb3d8eyJWIjoiMC4wLjAwMDAiLCJQIjoiV2luMzIiLCJBTiI6Ik1haWwiLCJXVCI6Mn0%3D%7C0%7C%7C%7C&sdata=AoKEN369CNyEwwKLaUJinZjTRFpXFj6IKZcMbOg8qew%3D&reserved=0 (accessed on 8 April 2024)).

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
