# Peer review of "Wild-Mouse-Derived Gut Microbiome Transplantation in Laboratory Mice Partly Alleviates House-Dust-Mite-Induced Allergic Airway Inflammation"

_microorganisms, 2024, doi:10.3390/microorganisms12122499_

Round 1

Reviewer 1 Report

Comments and Suggestions for Authors

With real interest, I read the manuscript microorganisms-3354281. Nicely written paper, with very elegantly presented results.

Comments:

1.        Lines 67-70. Indeed, asthma has many faces, also at the mechanistic level. However, other types of allergic airway inflammation mimicking human asthma can be modeled in mice as well, with HMD being a very good tool (PMID: 30267575).

2.        In this view, which type of allergic airway inflammation was targeted here. From what the Authors write and based on partly available data, one could assume that we could assume that it was type 2 allergic airway inflammation, right?

3.        In any case, it would be important to also show the numbers of BAL neutrophils.

4.        And here we come to the most important problem/limitation of this work. Actually, initially I was surprised that such an interesting, data-rich article was not published higher. From what I can see, two very important control groups seem to be missing, i.e. “SPF” and “Wild” mice, in whom no HDM-induced allergic airway inflammation was induced. Without those control groups, it is very difficult to state that allergic airway inflammation was successfully induced (and of course, what type of it got induced, if any). If the Authors do not have any data for such groups, the only thing they can do is stating that in their other works the same model worked well (reference/-s). Still, lack of the control groups need to be reported as a huge limitation and the data must be interpreted with great caution.

5.        Recently, the effect of intranasally delivered farm environment bacteria on gut microbiome and thus allergic airway inflammation has been reported (PMID: 36458896). How does this observation relate to your findings?

Author Response

With real interest, I read the manuscript microorganisms-3354281. Nicely written paper, with very elegantly presented results.

Comments:

  1. Lines 67-70. Indeed, asthma has many faces, also at the mechanistic level. However, other types of allergic airway inflammation mimicking human asthma can be modeled in mice as well, with HMD being a very good tool (PMID: 30267575).

AD1) Thank you for the relevant reference, which has now been included.

  1. In this view, which type of allergic airway inflammation was targeted here. From what the Authors write and based on partly available data, one could assume that we could assume that it was type 2 allergic airway inflammation, right?

AD2) This is indeed a relevant question. We have in house tested the type 2 inflammatory response in the lungs after different number of challenges, and found that this protocol was needed to properly induce a type 2 immune response compared to vehicle treated mice. This is however not yet published but is now mentioned in the manuscript. We do also observed an increase in neutrophils, and as such we would consider this a mixed model considering the above mentioned paper.

  1. In any case, it would be important to also show the numbers of BAL neutrophils.

AD3) We agree on the relevance of this and BAL neutrophil data as well as T and B cell numbers have now been included in figure 1 as suggested despite lack of significant differences.

  1. And here we come to the most important problem/limitation of this work. Actually, initially I was surprised that such an interesting, data-rich article was not published higher. From what I can see, two very important control groups seem to be missing, i.e. “SPF” and “Wild” mice, in whom no HDM-induced allergic airway inflammation was induced. Without those control groups, it is very difficult to state that allergic airway inflammation was successfully induced (and of course, what type of it got induced, if any). If the Authors do not have any data for such groups, the only thing they can do is stating that in their other works the same model worked well (reference/-s). Still, lack of the control groups need to be reported as a huge limitation and the data be interpreted with great caution.

Ad4) We agree that vehicle treated group could be beneficial e.g. for clarifying whether potential mechanisms such as changes in gut permeability measurements would be affected due to either a difference in disease state or directly by the microbiome differences. However, as the current study mainly aimed to investigate whether the disease state was affected by the wild microbiome, these groups were not included in order to reduce the number of mice used to only include those with a clear purpose. Healthy vehicle treated mice have no inflammatory cells in BAL or histology, so it would be a comparison to ‘zero’ (i.e. healthy unaffected tissue) for the primary outcomes if vehicle mice were included. Therefore they were not included, but IgE and cytokine measurements from vehicle treated mice could of course be useful to include as healthy levels are not ‘zero’. We have indeed previously shown that these are elevated in the model, but as the specific immune markers were unaffected by the microbiome in the current study, we found the veh groups irrelevant or unnecessary to conclude on the results. Nonetheless, we agree that this is of course a highly relevant notion, that has now been included as a limitation in the manuscript.

  1. Recently, the effect of intranasally delivered farm environment bacteria on gut microbiome and thus allergic airway inflammation has been reported (PMID: 36458896). How does this observation relate to your findings?

Ad5) Super relevant, thank you. This study only further supports the hygiene hypothesis similar to our results.

Reviewer 2 Report

Comments and Suggestions for Authors

The manuscript refers to Wild mouse-derived gut microbiome transplantation in laboratory mice, which partly alleviates house dust mite-induced allergic airway inflammation. The rationale of the article is adequate. The methodology is detailed and to the point. The results are well presented, and the Figures and Tables are adequate. The statistical analysis of the samples is correct. The results support the discussion. The only suggestion will probably be in the conclusions. The last part of the conclusions could be as a separate paragraph so that the main findings are separated from the second part, involving new research.

Author Response

The manuscript refers to Wild mouse-derived gut microbiome transplantation in laboratory mice, which partly alleviates house dust mite-induced allergic airway inflammation. The rationale of the article is adequate. The methodology is detailed and to the point. The results are well presented, and the Figures and Tables are adequate. The statistical analysis of the samples is correct. The results support the discussion. The only suggestion will probably be in the conclusions. The last part of the conclusions could be as a separate paragraph so that the main findings are separated from the second part, involving new research.

Thank you for your positive attitude towards our manuscript. The conclusions have now been separated into two parts as suggested.

Round 2

Reviewer 1 Report

Comments and Suggestions for Authors

Thanks for addressing my comments well. I have no further reservations.